# Depression and Functioning during the COVID-19 Pandemic among Adults across Tunisia

**DOI:** 10.3390/ijerph21101363

**Published:** 2024-10-16

**Authors:** Jessica E. Lambert, Fatma Charfi, Uta Ouali, Amina Aissa, Joop de Jong

**Affiliations:** 1DIGNITY, Danish Institute against Torture, 2100 Copenhagen, Denmark; 2Department of Child Psychiatry, Mongi Slim Hospital, La Marsa 8030, Tunisia; 3Razi Hospital La Manouba, Manouba 2010, Tunisia; 4Amsterdam University Medical Center, 1105 AZ Amsterdam, The Netherlands

**Keywords:** depression, Tunisia, COVID-19 pandemic, gender differences

## Abstract

We aimed to understand how risk (trauma history, health problems, financial problems, family problems) and protective (friend support, family support) factors influenced daily functioning (e.g., self-care, mobility, social participation) among Tunisian adults during the COVID-19 pandemic, directly and through their impact on depression, with a focus on gender differences. We recruited a representative sample of 2014 participants (1024 males, 990 females) using random digit dialing of mobile phone numbers across all 24 governorates in Tunisia. Females reported higher depression, greater impaired functioning, and a higher likelihood of having had COVID-19. Path analysis showed a good fit to the model when paths for males and females were allowed to vary, providing evidence for gender differences. Associations between trauma exposure and depression and depression and age with functioning were stronger among females. Social support from friends was a protective factor for males only. For males, all study variables were associated with functioning indirectly through their association with depression, except for support from friends. For females, family responsibilities and health problems had both direct and indirect effects, whereas other study variables were only linked with functioning through depression. Findings provide insights into factors that can be targeted in interventions aimed at reducing depression and improving daily functioning for males and females.

## 1. Introduction

Tunisia, a lower middle-income country in North Africa with a population of nearly 12 million, has undergone significant political and economic changes since the Arab Spring Revolution in 2011. Previously an authoritarian regime marked by corruption and human rights violations, Tunisia’s democratic transition brought freedom of speech and a stronger civil society. Although progress has been made in developing the mental health system [1], data on the prevalence of mental health issues remain scarce, especially among the general population. The most recent epidemiological study, conducted in 1995, found lifetime prevalence rates of 8% for depressive disorders and 1% for psychotic disorders [2]. These figures are likely outdated given the political instability, financial crises, and insecurity that have persisted since 2011, contributing to an increase in mental health problems [3]. Research points to rising rates of depression, post-traumatic stress disorder (PTSD), and nearly doubled suicide rates in the post-revolution period [3,4]. As with other countries around the world [5,6,7], the COVID-19 pandemic exacerbated the burden of mental health difficulties in the country, with women particularly affected [8,9,10].

In 2020, Tunisia initially managed the pandemic relatively well due to early lockdowns and strict measures [11]. By the end of the first year, the Ministry of Health had reported over 144,796 confirmed cases and 4896 deaths, with a case fatality rate that fluctuated between 2.5% and 3% during the early months [12,13]. However, the strain on the healthcare system grew as the virus spread, especially with new variants emerging in 2021. The second year saw a more severe impact, with the Delta variant contributing to a surge in cases and deaths during the summer of 2021, making Tunisia one of the most affected countries on the African continent [12]. By July 2021, Tunisia recorded over 200 deaths per day, and by August, total deaths had exceeded 20,000 [12,13]. The pandemic has also led to significant economic difficulties; as with other low- and middle-income countries (LMICs), insufficient assistance has been provided to those affected. Research in Tunisia during this period showed high rates of psychological distress among women [10], healthcare professionals [9], and individuals under quarantine [14]. Carta and colleagues [8] conducted an epidemiological study in La Manouba Governorate that demonstrated the pandemic was linked to an increased risk for depression, particularly among people struggling financially and women living in urban areas.

While informative, with the exception of the study by Carta and colleagues [8], most mental health research in Tunisia during and prior to the pandemic focused on specific populations with relatively small convenience samples, limiting their generalizability. There are substantial knowledge gaps on factors that promote and detract from well-being. Research that identifies risk and protective factors associated with psychosocial functioning, particularly during crisis situations [15], is important for understanding the needs of the population to inform relevant interventions. For example, health concerns, financial difficulties, and family problems surged due to the pandemic’s impact on the economy, health systems, and family dynamics during lockdowns. Individuals with a prior history of exposure to trauma were likely vulnerable during this time [14]. Simultaneously, social distancing measures and the general atmosphere of crisis increased the importance of social support, as reliance on family and friends became particularly salient for managing stress and maintaining mental health [15].

Understanding how stressors and protective factors during the pandemic differently affected males and females is also important, given that gender influences the prevalence, expression, and coping mechanisms for psychological distress and because these differences may be amplified in times of stress [16]. There is evidence that women were disproportionately affected by the pandemic globally [17], as the conditions exacerbated gender inequalities related to income [18] and family care duties [18,19,20] and contributed to an increase in partner violence [21]. Cross-sectional research across multiple countries [17] demonstrated a higher prevalence of mental health symptoms in women than in men.

Gender differences are particularly salient in contexts such as Tunisia, where socially prescribed gender roles—such as men working outside the home and generally spending more time in the community while women take care of the household—are common and deeply rooted in society. During the pandemic, due to restrictive issues across the country, men were required to spend more time in the home, with many losing their source of income. This contributed to family problems, including an increase in violence against women [10]. Investigating gender-specific pathways can provide insights into how interventions can be tailored to address the unique needs of men and women effectively.

In this study, we addressed knowledge gaps on the correlates of mental health and functioning of adults in Tunisia during the COVID-19 pandemic. The primary objective was to test a model that explains variation in depression and impaired functioning among adults. We focused on depression symptoms, given the evidence of an increase during the pandemic [8] and the fact that depression is the mental health difficulty most likely to contribute to impaired functioning [22]. Building on previous research in Tunisia [8] and other contexts [23,24,25], we selected risk factors likely to be salient during this time (trauma history, health concerns, financial difficulties, and difficulties with family responsibilities). We included social support from friends and social support from family as potential protective factors. In the model, we examined the role of risk and protective factors in predicting depression and, in turn, how these symptoms affect daily functioning. Specifically, we hypothesized that the associations between risk and protective factors and functioning would be mediated, at least in part, by depressive symptoms (see Figure 1). To provide a nuanced understanding, we also examined gender differences in these relationships, aiming to discern how associations might differ for males and females. Because the gender comparison was exploratory, we did not advance hypotheses about specific relations in the model; however, based on previous research, we anticipated females would report higher symptoms and more difficulties with functioning than males in the sample. This study contributes to the literature by simultaneously examining a set of risk and protective factors that, to our knowledge, have not been previously studied together in this context, and by recruiting a representative sample of adults from all governorates across Tunisia.

## 2. Methods

### 2.1. Participants

Data were collected from a total of 2014 adults (1024 males and 990 females). Females in the sample ranged in age from 18 to 95 (*M* = 40.27, *SD* = 15.02); 60.4% were married and 26.3% were employed. Regarding education level, 14.2% had no formal education, 24.6% completed primary education, 35.8% completed secondary school, 2.3% had a professional education, and 23% completed a university education. Males ranged in age from 18 to 85; 58.5% were married and 64.7% were employed. Regarding education level, 6.5% had no formal education, 28.1% completed primary education, 45.2% completed secondary education, 3.9% had a professional education, and 16.2% completed university.

### 2.2. Procedures

The study was approved by the Ethical Review Board of the Hospital Mongi Slim of La Marsa in Tunisia prior to data collection. Specifically, we submitted the full protocol, information about the research team, our timeline, and the research procedures to the board, who reviewed and gave approval. Ten experienced interviewers received a 4-h training workshop led by local psychiatrists. Participants reporting high levels of distress were provided with contact information for mental health resources in their localities.

Data were collected in June and July 2021 using a telephone survey method. Participants were randomly selected from lists provided by four telecom companies: Ooredoo Tunisie (43%), Tunisie Telecom (31.9%), Orange Tunisie (23.6%), and Lycamobile (1.5%). Phone numbers were randomly generated with equal probabilities, and quotas for sex, age, district, and governorate were controlled using a computer-assisted telephone interviewing system, thereby facilitating the identification of potential participants and the efficiency of the data collection process. Recruitment covered all 24 governorates, proportionally sampled based on the 2014 Population Census. The mobile telephony penetration rate was 124% as of September 2020, with national coverage exceeding 90% in both urban and rural areas.

A “5+” call design was implemented, making up to five attempts per number. If contact was made within these attempts, an additional five calls were made to reach the respondent. The overall response rate was 6.87%, which is typical for telephone interviews [26]. Participants who were reached were provided with a description of the study and asked to give verbal consent. Those consenting to take part in the study completed a 45-min telephone interview.

### 2.3. Measures

#### 2.3.1. Exposure to Potentially Traumatic Events and Stressors

Exposure to potentially traumatic events was assessed with a single item that read, “I would like to ask you if you have ever experienced or witnessed others experience very stressful life events. These stressful life events may include a severe car accident or other type of accident, emotional or physical violence from a spouse or romantic partner, being detained or tortured by police or authorities, witnessing a terrorist attack, or being a victim of other types of violence.” If participants responded “Yes”, they were asked what happened and when, with follow-up inquiries into the number of different events that occurred.

Exposure to daily stressors was assessed with items from the Brief Daily Stressors Screening Tool [27]. On this 10-item scale, respondents are asked to rate the degree to which they have been burdened by different stressors over the past 12 months on a 5-point scale ranging from 0 (not at all) to 4 (extremely). Each item includes a different potential stressor, including difficulties with family responsibilities, health problems, problems at work, problems with living situations, and problems with social relationships. In this study, we included three individual items that assessed the constructs of interest. Specifically, participants rated how much they have been burdened by health problems, financial problems, and difficulties with family responsibilities.

#### 2.3.2. Depression Symptoms

We used a version of the Patient Health Questionnaire-9 (PHQ-9) [28] that had previously been translated and validated in Tunisian Arabic [29]. The PHQ-9 is a widely used self-administered tool designed to screen for and measure the severity of depression symptoms. The PHQ-9 consists of nine items, each corresponding to one of the DSM-IV criteria for major depressive disorder. Respondents rate the frequency of symptoms such as anhedonia, sleep disturbances, and feelings of worthlessness over the past two weeks on a scale from 0 (not at all) to 3 (nearly every day). The total score ranges from 0 to 27, with higher scores indicating greater severity of depression. The PHQ-9 has demonstrated strong reliability and validity across diverse populations and is commonly used in both clinical and research settings to assess depressive symptoms and monitor treatment outcomes. In the validation study in Tunisia, the measure had Cronbach’s alpha = 0.84 with good criterion validity; a cutoff score of 10 on the measure had a sensitivity of 86.2 and specificity of 83.8 in detecting Major Depression [29]. In the present study, Cronbach’s Alpha was 0.80 for the full sample, 0.80 for males, and 0.79 for females.

#### 2.3.3. Impaired Daily Functioning

Impaired functioning was assessed using the World Health Organization Disability Assessment Schedule 2.0 (WHODAS 2.0) 12-item self-report scale [30]. This tool measures disability and functioning across six domains: cognition, mobility, self-care, getting along, life activities, and participation. Respondents rate their difficulty in performing activities over the past 30 days on a scale from 1 (none) to 5 (extreme or cannot do). The total score ranges from 12 to 60, with higher scores indicating greater impairment. The WHODAS 2.0 is widely validated and provides a comprehensive assessment of functional impairment applicable to various health conditions and populations. Tunisia was one of the countries involved in the initial development and validation of the measure led by the WHO [30]. Cronbach’s alpha was 0.80 for the full sample, 0.77 for males, and 0.82 for females.

#### 2.3.4. Social Support

Social support from friends and family was assessed using the Multidimensional Scale of Perceived Social Support (MSPSS) [31]. The MSPSS is a 12-item self-report scale that evaluates perceived social support from three sources: family, friends, and a significant other. Respondents rate each item on a 7-point Likert scale ranging from 1 (very strongly disagree) to 7 (very strongly agree). The scale provides subscale scores for each source of support and a total score, with higher scores indicating greater perceived support. An Arabic language version of the MSPSS has been validated with Lebanese adults [32,33]. Because approximately 40% of the participants were not married, in the present study, we used only the family support and friends support subscales. The Cronbach’s alpha for the 4-item family support scale was 0.75 (males = 0.74, females = 0.75). The alpha for the 4-item friend support was 0.83 (males = 0.83; females = 0.84).

#### 2.3.5. Demographic Information

A demographic questionnaire was created for the purpose of this study to assess the characteristics of the sample (age, marital status, employment status, education level). Because data were collected during the COVID-19 pandemic, we asked whether people had contracted the virus and if they were vaccinated.

#### 2.3.6. Translation and Pilot Testing

All measures except the demographic questionnaire and the Daily Stressors Screening Tool had previously been translated into Arabic. To ensure suitability for the Tunisian context and the vernacular local Arabic, the full protocol was back-translated by a professional interpreter and corrected by our bilingual research team. Pilot testing was conducted to ensure all items were understood.

### 2.4. Data Analysis Plan

Descriptive statistics and bivariate correlations were run in IBM SPSS version 28 (Armonk, NY, USA) [34]. For the main analysis, we employed a multi-group path analysis using IBM AMOS version 28 software [35] to test for potential gender differences in the model. Multigroup path analysis was chosen as the statistical method because it allows for the simultaneous estimation and comparison of models across different groups—in this case, males and females.

We set up the model such that there were direct paths from risk factors (health problems, family strain, financial problems, history of trauma), protective factors (family support, friend support), and the covariate age, to both depression and impaired functioning (See Figure 1). There was also a path from depression to impaired functioning. Exogenous variables that were significantly correlated were allowed to covary in the model.

A multi-group test allowed us to assess how well our data fit the proposed model using fit indices. We examined three model fit indices based on recommendations by Hu and Bentler [36]: the Goodness of Fit Index (GFI), Comparative Fit Index (CFI), and Root Mean Square Error of Approximation (RMSEA). These indices are commonly used to assess model fit, with GFI and CFI values closer to 1 indicating a good fit and RMSEA values below 0.06 suggesting an acceptable fit [37,38].

After the model setup was complete, it was fitted simultaneously to the two groups to test multi-group moderation based on participant gender. A chi-square difference test was used to compare test whether the overall model fit differently for males and females. We tested the mediation through bootstrapping with 5000 samples, estimating direct and indirect effects along with their confidence intervals. A series of Z-tests followed this to compare individual paths in the model [39]. We used *p* < 0.05 as the threshold for statistical significance.

## 3. Results

### 3.1. Preliminary Analyses

Descriptive statistics and statistical comparisons of scores for males and females are shown in Table 1. Females reported significantly higher stress related to health problems, higher depression scores, and higher scores on impaired functioning. Females also reported significantly higher support from family, but not friends. Males reported significantly more financial stress than females. There was no significant difference based on gender for family-related stress and support from friends. Depression scores were high for both genders; 30.5% of males and 34.5% of females scored above 10 on the PHQ-9. A score of 10 was found to be the threshold for clinically significant symptoms of depression among Tunisians [28]. Regarding COVID-19, 11.62% of males and 18.38% of females reported having had the virus, as confirmed by a test, at some time since the start of the pandemic. Males and females reported similar levels of vaccination: 13.76% and 15.25%, respectively.

Bivariate correlations between all variables are shown in Appendix A. Variables were related in expected ways, with stressors (lifetime trauma exposure, financial problems, health problems, and family problems) positively correlating with depression symptoms, and impaired functioning and support from friends and family negatively correlating with these variables. Having had COVID-19 and vaccination status were only marginally related to study variables and, as such, were excluded from the main analysis.

### 3.2. Main Analyses

#### 3.2.1. Multigroup Analysis Testing for Gender Differences

The unconstrained model, where paths were allowed to vary between males and females, presented an excellent fit to the data χ^2^ (10, *N* = 2014) = 22.31, *p* = 0.014 (GFI = 0.99, CFI = 0.99, RMSEA = 0.02). In the fully constrained model, where all parameters were set to be equal across groups, there was a notable decline in model fit, χ^2^ (59, *N* = 2014) = 251.88, *p* < 0.001, with fit indices indicating a less-than-optimal fit (GFI = 0.92, CFI = 0.89, RMSEA = 0.04) compared to the unconstrained model. We conducted a chi-square difference test to compare the fit of the constrained model against the unconstrained model, resulting in Δχ^2^ = 229.57 with a change in degrees of freedom of Δdf = 49, which was statistically significant, *p* < 0.001. This significant result from the chi-square difference test indicates that the constraints imposed on the model (i.e., the perimeters being equal across groups) significantly worsened the model fit. This finding suggests that the paths in the model differ between males and females. As such, path coefficients were examined separately for males and females (See Table 2). We conducted Z-tests to test the differences in the individual paths between males and females.

As shown in Table 2, although the models differed, there were few paths in the model with estimates that were significantly different based on gender. Regarding risk factors for depression, for both males and females, older age was associated with lower depression, while more problems with family responsibilities, health issues, and financial difficulties were associated with higher depression. The one exception was exposure to trauma; whereas having experienced a traumatic event was associated with higher depression for both genders, this association was significantly stronger for females. For protective factors, social support from friends was associated with lower depression for males, but not females, although the difference between the two estimates was not statistically significant. The full model accounted for 28% of the variance in depression for males and 27% of the variance for females.

Regarding risk factors for impaired functioning, depression was significantly associated with impaired functioning for males and females; however, the association was significantly stronger for females, meaning that, for these individuals, depression symptoms explained significantly more of the variation in functioning than for males. Age was also more strongly associated with impaired functioning for females than for males, with older females having more impaired functioning. Family and health problems were directly associated with imparted functioning, whereas trauma exposure and financial problems did not have direct associations. These paths were not significantly different for males and females.

For hypothesized protective factors, only support from friends was directly and inversely associated with impaired functioning for males. However, the estimates were not statistically significantly different for males and females. The model accounted for 32% of the variance in impaired functioning for males and 33% for females.

#### 3.2.2. Mediation Results

The indirect effects are shown in Table 3. For males in the sample, problems with family responsibilities, health problems, and support from friends were significantly associated with impaired daily functioning, both directly and indirectly, through depression. Trauma exposure and financial problems were associated with impaired functioning only indirectly through their association with increased depression. Support from friends was associated directly with depression and functioning, while the indirect effect was non-significant (*p* = 0.06).

For females in the sample, problems with family responsibilities and health problems were associated with impaired functioning, both directly and indirectly, through their influence on depression. Trauma exposure, financial problems, and support from family were only significantly associated with functioning indirectly through their association with depression. Support from friends was not significantly associated with depression or functioning.

## 4. Discussion

In this study, we tested a model to understand variation in daily functioning among adults in Tunisia during the COVID-19 pandemic. Specifically, we evaluated the indirect associations of risk and protective factors on impaired functioning through depression symptoms, with a focus on gender differences. Consistent with other studies on mental health during the pandemic in Tunisia [8,9,10,14] and other countries [5,6,7], participants reported high levels of psychological distress, with more than 30% reporting depression scores over the clinical cutoff. In the preliminary analysis, we found that females in the sample reported a slightly higher rate of having had COVID-19 (18.38% vs. 11.62%), despite being vaccinated at approximately the same rate as men. Other research suggests rates of COVID-19 infection rates in Tunisia have been similar between genders [40]; however, we were unable to locate information on differences in vaccination rates for males and females. Having had the virus or vaccination status was not meaningfully related to study variables. Our hypothesized model fit the data well when we allowed paths to vary between males and females, providing evidence that the model significantly differed for males and females. Overall, the findings demonstrate the salience of a lifetime history of exposure to potentially traumatic events, as well as current stress related to health concerns, financial difficulties, and family responsibilities, in understanding elevated depression and impaired functioning among the population of adults during the pandemic. In some cases, supportive relationships were an important protective factor. The results highlighted potentially important differences and similarities based on gender, which are elaborated below.

In the model, older participants reported lower depression scores but more impaired daily functioning. The mediation results were similar for males and females in the model. Stress due to family responsibilities and health problems had both direct and indirect effects on functioning through depression. Trauma exposure, financial problems, and family support were only indirectly associated with impaired functioning through their association with depression. Taken together, this means that certain stressors (trauma, financial problems) are likely to contribute to depression symptoms, which, in turn, lead to more impaired functioning, whereas other stressors (health problems, family responsibilities) also contribute to impaired functioning apart from their association with depression. Furthermore, the association between family support and functioning was mediated by depression, such that better support was associated with lower depression, and, in turn, better functioning. This finding highlights the importance of family networks in the population.

Notable gender differences also emerged in the findings. In the preliminary analysis, we found females reported significantly more stress related to health problems than males. Similar to the research on gender differences globally [41], women reported higher depressive symptoms and more impaired functioning. Women also reported significantly more family support than men, which is consistent with cultural norms in Tunisia, where many women center their lives around extended family networks. Men were significantly more likely to report a lifetime history of a potentially traumatic event (25.97% vs. 20.20%), although it is not possible to determine if this was due to actual prevalence or reporting bias.

In the test of the model, although men were more likely to report having experienced a potentially traumatic event at some point in their lives, trauma history was more strongly correlated with depression symptoms for women. This is consistent with research on the impacts of traumatic events globally, with females generally having a higher risk of psychological problems following traumatic events [42]. This could be due to the nature of the types of events to which women are exposed, differences in coping, or biological differences [43]. For example, women are more likely to experience sexual violence, which has a higher probability of leading to psychological difficulties than other potentially traumatic events [44]. In the present sample, we were not able to draw conclusions based on different types of events, because around 30% of the sample declined to share details on what kind of event they experienced.

Age was more strongly related to impaired functioning for women than for men. This could be due to the higher stress related to health problems we found in our study or to the changing social roles of aging women in this context. For example, Delanoe and colleagues [45] found that Tunisian women aged 45 to 70 reported a sense of social degradation linked with being in menopause.

Depression was more strongly associated with impaired functioning for women than for men. One explanation for this is that the traditional role of men as the providers and heads of household in this context leads to a pattern of behavior where men continue to function despite being depressed. Alternatively, Kamel and Hentati [46] found that men in Tunisia reported relatively more resilience during the first lockdown of the pandemic, so it is also possible that men were generally coping better with adversity during this time. The time during the lockdown was particularly stressful for women globally, with research showing a higher burden related to family care and employment instability [47,48,49]. Although not reported by participants in this study, Tunisia has high rates of violence against women, which increased during the pandemic [10], which could have influenced findings.

The final gender difference was related to the role of social support from friends. Specifically, for males, social support from friends was significantly associated with both lower levels of depression and better overall functioning. In contrast, for females in the sample, social support from friends did not show a significant relationship with either depression or functioning. This discrepancy may be influenced by cultural factors. In Tunisia, women often experience lower levels of social integration and spend more time within the home, potentially reducing the salience and impact of support from friends outside the family. As a result, women may rely more on family members for emotional and practical support during times of stress, rather than on friends.

It is important to note that cultural norms in Tunisia may have significantly influenced both the data collection process and our aforementioned results. In Tunisian society, mental health issues often carry stigma, and open discussions about psychological distress are limited. This stigma may have led participants to underreport symptoms of depression or functional impairments due to fear of social judgment or repercussions. Gender roles are deeply ingrained, with expectations that men exhibit resilience and refrain from expressing vulnerability. Consequently, men might have underreported emotional difficulties to align with societal expectations of masculinity. Women, on the other hand, might have been more willing to report emotional distress but less inclined to disclose family-related problems or experiences of domestic violence, aiming to preserve family honor and privacy.

These cultural factors also affect the interpretation of our findings. The higher reported rates of depressive symptoms among women could reflect both genuine disparities in mental health experiences and differences in reporting behaviors influenced by cultural norms. The importance of family and community in Tunisian culture may amplify the role of social support as a protective factor, but it could also intensify stressors related to family responsibilities, especially for women.

Key findings must be interpreted considering the study’s limitations. Collecting data during the pandemic may have resulted in higher levels of daily stress and depression. Additionally, aspects of the methodology could have influenced the results. Conducting interviews over the phone allowed us to reach a wider sample during a time when face-to-face interviews would have been challenging because of the pandemic; nevertheless, this method could have biased the findings. Our response rate was low, although not atypical for a telephone survey [26]. The use of telephone interviewing potentially influenced the degree to which participants felt comfortable disclosing sensitive information. This could be the reason why relatively few participants reported having experienced a potentially traumatic event. In comparison, a large community-based study of adults in Eastern Central Tunisia found that approximately one-third of women and nearly half of men had experienced some form of physical abuse in childhood and a similar percentage had been exposed to community violence [50]. Actual rates of trauma exposure for women may be even higher; according to recent data from the Ministry of Women, nearly 47% of women in Tunisia have experienced some form of partner violence in their lifetime, and the prevalence is believed to have increased during the pandemic [51].

Distress was assessed with symptom scales rather than diagnostic interviews, which can result in overestimates of the actual prevalence of disorders. Measures were translated from English to Arabic, some for previous studies and some for this project. Although the protocol was evaluated by our bilingual team and piloted, it is possible that important culturally specific terminology was omitted, thereby compromising the validity of the findings.

Despite the limitations, this study contributes to the growing body of work focused on understanding mental health in the Tunisian context and findings point to important areas for future research. Future studies that include diagnostic measures to clarify the prevalence of common mental disorders are needed to inform mental health policy and practice in this context. Research focused on the role of social support and how different social ties function for males and females is important for informing psychosocial interventions aimed at buffering daily stress.

## 5. Conclusions

This study provides valuable insights into the factors contributing to impaired daily functioning among adults in Tunisia during the COVID-19 pandemic. Overall, findings point to the need for mental health interventions that target depression, as well as programming aimed at helping people cope with notable stressors. The findings highlight the importance of considering lifetime exposure to trauma, particularly for women, along with ongoing stress from family issues, financial difficulties, and health problems, in understanding variability in depression, which, in turn, is linked to impaired functioning. Future research should continue to explore these dynamics, particularly in light of the limitations noted, to develop culturally appropriate strategies that enhance resilience and support within this population.

## Figures and Tables

**Figure 1 ijerph-21-01363-f001:**
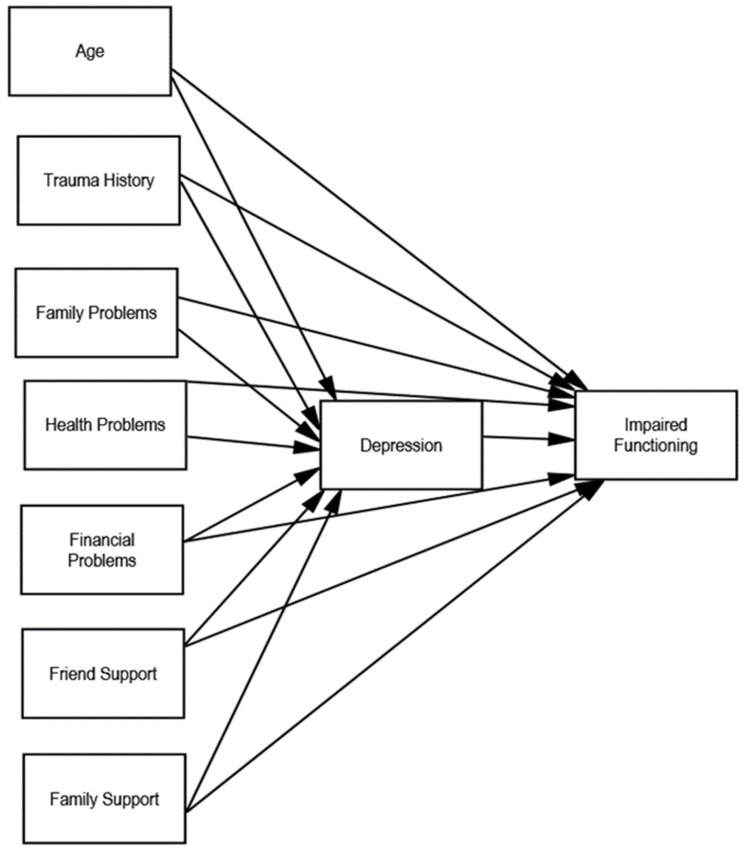
Hypothesized mediation model.

**Table 1 ijerph-21-01363-t001:** Descriptive statistics and gender comparison on study variables.

	Males	Females		
Variable	*N* (%Yes)	*N* (%Yes)	χ^2^ (DF)	*p*-value
COVID-19	119 (11.62%)	182 (18.38%)	18.11 (1)	<0.001
Vaccinated	141 (13.76%)	151 (15.25%)	0.89 (1)	0.376
Trauma	266 (25.97%)	200 (20.20%)	9.44 (1)	<0.001
	***M* (*SD*)**	***M* (*SD*)**	** *t* ** **-value (df)**	** *p* ** **-value**
Age	40.27 (15.02)	40.81 (15.36)	−0.78 (2012)	0.433
Family Problems	2.02 (1.65)	2.15 (1.64)	−1.83 (2012)	0.068
Health Problems	1.00 (1.43)	1.49 (1.62)	−7.23 (2012)	<0.001
Financial Problems	2.51 (1.61)	2.36 (1.60)	2.17 (2012)	0.03
Friend Support	4.69 (2.15)	4.88 (2.18)	−1.82 (1852)	0.068
Family Support	5.63 (1.73)	5.87 (1.64)	−3.15 (1993)	0.002
Depression	8.46 (6.31)	9.32 (6.11)	−3.10 (2012)	0.002
Impaired Functioning	19.12 (7.30)	21.67 (8.88)	−7.07 (2012)	<0.001

**Table 2 ijerph-21-01363-t002:** Path coefficients.

	Males		Females		
	Estimate (SE)	*p*	Estimate (SE)	*p*	Z-Score
Depression					
Age	−0.049	<0.001	−0.038	<0.001	0.642
Trauma exposure	1.106	0.004	3.394	<0.001	4.015 ***
Family problems	0.859	<0.001	0.663	<0.001	−1.136
Health problems	0.897	<0.001	0.799	<0.001	−0.556
Financial problems	0.806	<0.001	0.580	<0.001	−1.273
Friend support	−0.178	0.031	−0.115	0.154	0.549
Family support	−0.377	<0.001	−0.348	<0.001	0.196
Impaired Functioning					
Depression	0.518	<0.001	0.645	<0.001	2.158 **
Age	0.025	0.049	0.079	<0.001	2.557 **
Trauma exposure	0.348	0.421	0.468	0.435	0.162
Family problems	0.275	0.048	0.388	0.023	0.511
Health problems	0.738	<0.001	0.735	0.001	−0.016
Financial problems	−0.036	0.800	−0.148	0.398	−0.494
Friend support	−0.238	0.010	−0.163	0.139	0.520
Family support	−0.152	0.186	−0.145	0.332	0.039

** *p* < 0.01, *** *p* < 0.001.

**Table 3 ijerph-21-01363-t003:** Indirect effects.

	Males		Females	
Indirect Effect	Estimate (SE)	95% CI	Estimate (SE)	95% CI
Trauma → Depression → Functioning	0.57 (0.22)	[0.17, 0.99]	1.19 (0.39)	[1.57, 2.90]
Family problems → Depression → Functioning	0.44 (0.08)	[0.31, 0.60]	0.43 (0.09)	[0.27, 0.62]
Health problems → Depression → Functioning	0.46 (0.08)	[0.31, 0.64]	0.52 (0.10)	[0.33, 0.72]
Financial problems → Depression → Functioning	0.41 (0.08)	[0.28, 0.58]	0.37 (0.09)	[0.21, 0.56]
Friend support → Depression → Functioning	−0.09 (0.05)	[−19, 0.01]	−0.07 (0.06)	[−0.19, 0.03]
Family support → Depression → Functioning	−0.20 (0.60)	[−0.32, −0.09]	−0.23 (0.08)	[−0.40, −0.07]

Note. Estimates are unstandardized. Standard errors and 95% confidence intervals were bootstrapped using 5000 samples.

## Data Availability

Data are available on request from the first author.

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
