# Peer review of "Depression and Functioning during the COVID-19 Pandemic among Adults across Tunisia"

_ijerph, 2024, doi:10.3390/ijerph21101363_

Round 1
Reviewer 1 Report
Comments and Suggestions for Authors
After reviewing the paper, I would like to suggest several improvements that could enhance the clarity and impact of the research.
Abstract
It is essential to define and measure the term "impaired functioning" more clearly within the abstract and give an explanation of how the results inform public health strategies or mental health interventions in Tunisia.
Introduction
The introduction would benefit from a discussion on how gender roles influenced mental health during the pandemic.
Methods
I recommend adding details about the diversity of participants in terms of age, education, and marital status to ensure that the sample accurately reflects the population of Tunisia. Furthermore, it is important to provide more information about the ethical approval process to enhance transparency. The effectiveness of the recruitment strategy should also be discussed, particularly in light of the lower response rate observed in the study.
Results
Give a brief explanation of the indices (GFI, CFI, RMSEA) used in the analysis would be beneficial. It would also be helpful to state the threshold for significance that was employed in this study to clarify the criteria for interpreting the results.
Discussion
The discussion should include an exploration of how cultural norms may affect both the data collection process and the interpretation of results.
Comments on the Quality of English LanguageModerate editing of English language required.
Author Response
Thank you for taking the time to review our manuscript. We have addressed each comment below, and uploaded the new draft with tracked changes.
Comment 1: It is essential to define and measure the term "impaired functioning" more clearly within the abstract and give an explanation of how the results inform public health strategies or mental health interventions in Tunisia.
Response 1: These are important clarifications. Given the word limit for the Abstract we have briefly included a definition of how impaired functioning was measured in the first sentence “We aimed to understand how risk (trauma history, health concerns, financial problems, family problems) and protective (friend support, family support) factors influenced daily functioning (e.g., self-care, mobility, social participation) among Tunisian adults during the COVID-19 pandemic, directly and through their impact on depression, with a focus on gender differences.” On the second point, we changed the last sentence to read: “Findings provide insights into factors that can be targeted in interventions aimed at reducing depression and improving daily functioning for males and females.”
Comment 2: The introduction would benefit from a discussion on how gender roles influenced mental health during the pandemic.
Response 2: Thanks for this feedback. We have incorporated a brief discussion on gender in the pandemic and edited our previous version to make the rationale for focusing on gender clearer. The paras on lines 72 to 87 now read:
Understanding how stressors and protective factors during the pandemic differently affected males and females is also important, given that gender influences the prevalence, expression, and coping mechanisms for psychological distress, and these differences may be amplified in times of stress [16]. There is evidence that women were disproportionately affected by the pandemic globally [17], as the conditions exacerbated gender inequalities related to income [18], family care duties [18-20], and contributed to an in increase in partner violence [21]. Cross-sectional research across multiple countries [17] demonstrated higher prevalence of mental health symptoms in women than in men.
Gender differences are particularly salient in contexts like Tunisia, where socially prescribed gender roles--such as men working outside the home and generally spending more time in the community while women take care of the household-- are common and deeply rooted in the society. During the pandemic, due to restrictive issues across the country, men were required to spend more time in the home, many losing their source of income. This contributed to family problems including an increase in violence against women [10]. Investigating gender-specific pathways can provide insights into how interventions can be tailored to address the unique needs of men and women effectively.
Comment 3: I recommend adding details about the diversity of participants in terms of age, education, and marital status to ensure that the sample accurately reflects the population of Tunisia.
Response 3: This information was already included under the section 2.1 Participants, kindly inform us if more detail is required:
Data were collected from a total of 2014 adults, (1024 males and 990 female). Females in the sample ranged in age from 18 to 95 (M = 40.27, SD = 15.02), 60.4% were married, and 26.3% employed. Regarding education level, 14.2% had no formal education, 24.6% completed primary education, 35.8% completed secondary school, 2.3% had a professional education, and 23% completed a university education. Males ranged in age from 18 to 85, 58.5% were married, 64.7% were employed. Regarding education level, 6.5% had no for-mal education, 28.1% completed primary education, 45.2 completed secondary education, 3.9% had a professional education and 16.2% completed university.
Response 4: Furthermore, it is important to provide more information about the ethical approval process to enhance transparency.
In section 2.1 we included that the study was approved by the Ethical Review Board of the Hospital Mongi Slim of La Marsa in Tunisia prior to data collection. Specifically, we submitted the full protocol, information about the research team, our timeline, and the research procedures to the board who reviewed and gave approval. Kindly inform us if more details are required.
Comment 5: The effectiveness of the recruitment strategy should also be discussed, particularly in light of the lower response rate observed in the study.
Response 5: This is included as a limitation in the discussion. Specifically: “Conducting interviews over the phone allowed us to reach a wider sample during a time when face-to-face interviews would have been challenging because of the pandemic; nevertheless, this method could have biased the findings. Our response rate was low, although not atypical for a telephone survey [26]. The use of telephone interviewing potentially influenced the degree to which participants felt comfortable disclosing sensitive information.”
Comment 6: Give a brief explanation of the indices (GFI, CFI, RMSEA) used in the analysis would be beneficial.
Response 6: We clarified this in section 2.4 Data Analysis Plan, including citations for why these were selected:
Descriptive statistics and bivariate correlations were run in SPSS version 28 [34]. For the main analysis, we employed a multi-group path analysis using AMOS version 28 software [35] to test for potential gender differences in the model. Multigroup path analysis was chosen as the statistical method because it allows for the simultaneous estimation and comparison of models across different groups—in this case, males and females.
We set up the model such that there were direct paths from risk factors (health problems, family strain, financial problems, history of trauma), protective factors (family sup-port, friend support), and the covariate age, to both depression and impaired functioning (See Figure 1). There was also a path from depression to impaired functioning. Exogenous variables that were significantly correlated were allowed to covary in the model.
A multi-group test allowed us to assess how well our data fit the proposed model using fit indices. We examined three model fit indices based on recommendations by Hu and Bentler [36]: the Goodness of Fit Index (GFI), Comparative Fit Index (CFI), and Root Mean Square Error of Approximation (RMSEA). These indices are commonly used to assess model fit, with GFI and CFI values closer to 1 indicating a good fit, and RMSEA values below 0.06 suggesting an acceptable fit [37, 38].
After the model setup was complete, it was fitted simultaneously to the two groups to test multi-group moderation based on participant gender. A chi-square difference test was used to compare test whether the overall model fit differently for males and females. We tested the mediation through bootstrapping with 5000 samples, estimating direct and in-direct effects along with their confidence intervals. A series of Z-tests followed this to compare individual paths in the model [39]. We used p < .05 as the threshold for statistical significance.
Comment 7: It would also be helpful to state the threshold for significance that was employed in this study to clarify the criteria for interpreting the results.
Response 7: As shown above we included this sentence on line 235: We used p < .05 as the threshold for statistical significance.
Comment 8: The discussion should include an exploration of how cultural norms may affect both the data collection process and the interpretation of results.
Response 8: This is an excellent recommendation. We added the following to the discussion on lines 397-413:
It is important to note that cultural norms in Tunisia may have significantly influenced both the data collection process and our aforementioned results. In Tunisian society, mental health issues often carry stigma, and open discussions about psychological distress are limited. This stigma may have led participants to underreport symptoms of depression or functional impairments due to fear of social judgment or repercussions. Gender roles are deeply ingrained, with expectations that men exhibit resilience and refrain from expressing vulnerability. Consequently, men might have underreported emotional difficulties to align with societal expectations of masculinity. Women, on the other hand, might have been more willing to report emotional distress but less inclined to disclose family-related problems or experiences of domestic violence, aiming to preserve family honor and privacy.
These cultural factors also affect the interpretation of our findings. The higher reported rates of depressive symptoms among women could reflect both genuine disparities in mental health experiences and differences in reporting behaviors influenced by cultural norms. The importance of family and community in Tunisian culture may amplify the role of social support as a protective factor, but it could also intensify stressors related to family responsibilities, especially for women.
Reviewer 2 Report
Comments and Suggestions for Authors
This article discusses depression and functioning during COVID-19 among Tunisian adults.
Below are my observations for improving the value of this research for the readers.
Introduction
- The first paragraph seems to focus more on self-citation. While the information about training mental health workers is valuable, it could be more appropriate here to discuss the burden of depression and impaired functioning.
- In the introduction, provide more detailed information regarding COVID-19 morbidity and mortality in the country during the first and second years of the pandemic. This will help illustrate the burden of the pandemic more comprehensively.
- Clearly state how the present study builds upon or diverges from previous research presented in the introduction. Additionally, explicitly identify the research gap that this study aims to address.
Methods
4. Specify the measures that were implemented to reduce interviewer bias. As a careful reader, it appears that conducting 2014 interviews, each lasting 45 minutes, with a response rate of 6.87% over a two-month period might be quite challenging.
- Provide a more detailed explanation of why multigroup path analysis was chosen as the statistical method, especially for readers who may be less familiar with AMOS.
Results
6. In line 231, there is an incomplete sentence. Please revise it to ensure the message is conveyed clearly.
Discussion
7. In the discussion, you mention that more females had COVID-19 compared to males. Expand on this by discussing whether your findings on morbidity and vaccination rates are consistent with national COVID-19 statistics.
Comments on the Quality of English LanguageNo issues with the quality of English Language were detected.
Author Response
Many thanks for your thoughtful feedback. We have addressed each of you comments below, and uploaded a revised manuscript with track changes.
Comment 1: The first paragraph seems to focus more on self-citation. While the information about training mental health workers is valuable, it could be more appropriate here to discuss the burden of depression and impaired functioning.
Response 1: Fair point, we combined the first two paragraphs, deleting some specifics about MHGAP, but maintaining sufficient background information for an international readership who may be unfamiliar with the history and context of Tunisia. The revised first para now reads:
Tunisia, a lower middle-income country in North Africa with a population of nearly 12 million, has undergone significant political and economic changes since the Arab Spring Revolution in 2011. Previously an authoritarian regime marked by corruption and human rights violations, Tunisia's democratic transition brought freedom of speech and a stronger civil society. Although progress has been made in developing the mental health system [1], data on the prevalence of mental health issues remain scarce, especially among the general population. The most recent epidemiological study, conducted in 1995, found lifetime prevalence rates of 8% for depressive disorders and 1% for psychotic disorders [4]. These figures are likely outdated given the political instability, financial crises, and insecurity that have persisted since 2011, contributing to an increase in mental health problems [5]. Recent research points to rising rates of depression, PTSD, and nearly doubled suicide rates in the post-revolution period [5,6]. Like in other countries around the world, the COVID-19 pandemic has further exacerbated the burden of mental health difficulties in the country [7–9].
Comment 2: In the introduction, provide more detailed information regarding COVID-19 morbidity and mortality in the country during the first and second years of the pandemic. This will help illustrate the burden of the pandemic more comprehensively.
Response 2: Good suggestion we added a para on the pandemic during 2020 and 2021, lines 43 to 58:
In 2020, Tunisia initially managed the pandemic relatively well due to early lock-downs and strict measures [11]. By the end of the first year, the Ministry of Health had re-ported over 144,796 confirmed cases and 4,896 deaths, with a case fatality rate that fluctuated between 2.5% and 3% during the early months [12, 13]. However, the strain on the healthcare system grew as the virus spread, especially with new variants emerging in 2021. The second year saw a more severe impact, with the Delta variant contributing to a surge in cases and deaths during the summer of 2021, making Tunisia one of the most affected countries on the African continent [12]. By July 2021, Tunisia recorded over 200 deaths per day, and by August, total deaths had exceeded 20,000 [12][13]. The pandemic has also led to significant economic difficulties; like other low- and middle-income countries (LMICs), insufficient assistance has been provided to those affected. Research in Tunisia during this period showed high rates of psychological distress among women [10], healthcare professionals [9], and individuals under quarantine [14]. Carta and colleagues [8] conducted an epidemiological study in La Manouba Governorate, that demonstrated the pandemic was linked to an increased risk for depression, particularly among people struggling financially, and women living in urban areas.
Comment 3: Clearly state how the present study builds upon or diverges from previous research presented in the introduction. Additionally, explicitly identify the research gap that this study aims to address.
Response 3: We edited the para beginning line 59, and the para beginning line 80 of the introduction to more clearly state that we are addressing a gap in the literature on risk and protective factors associated with mental health and functioning among adults. We conclude the Introduction with this sentence: “This study contributes to the literature by simultaneously examining a set of risk and protective factors that, to our knowledge, have not been previously studied together in this context, and by recruiting a representative sample of adults from all governorates across Tunisia.” (lines 105-108)
Comment 4: Specify the measures that were implemented to reduce interviewer bias. As a careful reader, it appears that conducting 2014 interviews, each lasting 45 minutes, with a response rate of 6.87% over a two-month period might be quite challenging.
Response 4: We clarified in 2.2 that there were 10 interviewers who used a computer assisted telephone interviewing system that facilitated the calling process, allowing efficiency in reaching the high number of participants.
Comment 5: Provide a more detailed explanation of why multigroup path analysis was chosen as the statistical method, especially for readers who may be less familiar with AMOS.
Response 5: We rewrote section 2.4 to more clearly explain our analysis, it now reads:
Descriptive statistics and bivariate correlations were run in SPSS version 28 [34]. For the main analysis, we employed a multi-group path analysis using AMOS version 28 software [35] to test for potential gender differences in the model. Multigroup path analysis was chosen as the statistical method because it allows for the simultaneous estimation and comparison of models across different groups—in this case, males and females.
We set up the model such that there were direct paths from risk factors (health problems, family strain, financial problems, history of trauma), protective factors (family support, friend support), and the covariate age, to both depression and impaired functioning (See Figure 1). There was also a path from depression to impaired functioning. Exogenous variables that were significantly correlated were allowed to covary in the model.
A multi-group test allowed us to assess how well our data fit the proposed model using fit indices. We examined three model fit indices based on recommendations by Hu and Bentler [36]: the Goodness of Fit Index (GFI), Comparative Fit Index (CFI), and Root Mean Square Error of Approximation (RMSEA). These indices are commonly used to assess model fit, with GFI and CFI values closer to 1 indicating a good fit, and RMSEA values below 0.06 suggesting an acceptable fit [37, 38].
After the model setup was complete, it was fitted simultaneously to the two groups to test multi-group moderation based on participant gender. A chi-square difference test was used to compare test whether the overall model fit differently for males and females. We tested the mediation through bootstrapping with 5000 samples, estimating direct and indirect effects along with their confidence intervals. A series of Z-tests followed this to compare individual paths in the model [39]. We used p < .05 as the threshold for statistical significance.
Comment 6: In line 231, there is an incomplete sentence. Please revise it to ensure the message is conveyed clearly.
Response 6: Thanks for catching this error, the line now reads (beginning on line 245): Depression scores were high for both genders; 30.5% of males and 34.5% of females scored above 10 on the PHQ-9. A score of 10 was found to be the threshold for clinically significant symptoms of depression among Tunisians [28].
Comment 7: In the discussion, you mention that more females had COVID-19 compared to males. Expand on this by discussing whether your findings on morbidity and vaccination rates are consistent with national COVID-19 statistics.
Response 7: Our findings are somewhat similar, usually studies have found rates of infection to be similar for males and females. We were not able to find information on vaccination status disaggregated by gender. We added this sentence to the first para of the discussion: Other research suggests rates of COVID-19 infection rates in Tunisia have been similar between genders [40]; however, we were unable to locate information on differences in vaccination rates for males and females.
Round 2
Reviewer 1 Report
Comments and Suggestions for Authors
This version has been improved and accepted in this form.